# Functional Analysis of the *PCCA* and *PCCB* Gene Variants Predicted to Affect Splicing

**DOI:** 10.3390/ijms22084154

**Published:** 2021-04-16

**Authors:** Igor Bychkov, Artur Galushkin, Alexandra Filatova, Andrey Nekrasov, Marina Kurkina, Galina Baydakova, Alexandra Ilyushkina, Mikhail Skoblov, Ekaterina Zakharova

**Affiliations:** Research Centre for Medical Genetics, 115522 Moscow, Russia; arthurgaluschkin@gmail.com (A.G.); maacc@yandex.ru (A.F.); aroneru@gmail.com (A.N.); kurkina_marina87@mail.ru (M.K.); gb2003@yandex.ru (G.B.); alexilyuskina@gmail.com (A.I.); mskoblov@gmail.com (M.S.); doctor.zakharova@gmail.com (E.Z.)

**Keywords:** splicing variants, variant classification, functional analysis, *PCCA* and *PCCB*

## Abstract

It is estimated that up to one-third of all variants causing inherited diseases affect splicing; however, their deleterious effects and roles in disease pathogenesis are often not fully characterized. Given their prevalence and the development of various antisense-based splice-modulating approaches, pathogenic splicing variants have become an important object of genomic medicine. To improve the accuracy of variant interpretation in public mutation repositories, we applied the minigene splicing assay to study the effects of 24 variants that were predicted to affect normal splicing in the genes associated with propionic acidemia (PA)—*PCCA* and *PCCB*. As a result, 13 variants (including one missense and two synonymous variants) demonstrated a significant alteration of splicing with the predicted deleterious effect at the protein level and were characterized as spliceogenic loss-of-function variants. The analysis of the available data for the studied variants and application of the American College of Medical Genetics and the Association for Molecular Pathology (ACMG/AMP) guidelines allowed us to precisely classify five of the variants and change the pathogenic status of nine. Using the example of the PA genes, we demonstrated the utility of the minigene splicing assay in the fast and effective assessment of the spliceogenic effect for identified variants and highlight the necessity of their standardized classification.

## 1. Introduction

Pathogenic variants in the *PCCA* and *PCCB* genes are responsible for the rare autosomal recessive metabolic disease called propionic acidemia (PA) (OMIM#606054). The products of these genes form the heterododecameric enzyme propionyl CoA carboxylase (PCC), which converts propionyl CoA to methymalonyl CoA in the mitochondrial matrix [1]. The deficient activity of PCC leads to accumulation of toxic propionic acid metabolites and dysfunction of the respiratory chain and the urea cycle pathway. PA is clinically heterogeneous and has a variable age of onset ranging from severe neonatal forms to mild adult-onset forms [2]. In most cases, the symptoms of PA manifest in the early neonatal period and without treatment quickly become life-threatening. The symptoms include seizures, poor feeding, vomiting, hypotonia, metabolic acidosis, ketonuria, hypoglycemia, hyperammonemia and cytopenia [3]. Timely medical and dietary interventions could significantly improve the patient’s condition and stabilize the metabolic state. Therefore, fast and proper genetic diagnosis can play an important role in patient management [3,4,5]. 

The ClinVar (https://www.ncbi.nlm.nih.gov/clinvar) (accessed on 15 April 2021) and Human Gene Mutation (http://www.hgmd.cf.ac.uk/) (accessed on 15 April 2021) databases are the main repositories of PA variants and currently comprise 720 and 322 unique variants, respectively. Missense variants are the predominant cause of PA, followed by small insertions and deletions, splicing variants and large genomic deletions [6]. Due to the increasing availability of molecular genetic diagnostics, the rapidly growing number of novel variants presents a challenge to researchers to establish their molecular consequence. In the case of missense variants, enzymatic assay and Western blot are widely used to reveal their deleterious effect on the PCC structure and activity [7,8]. 

Another class of variants that disrupts normal splicing is of particular interest due to development of the specific splice-modulating approaches for PA [9,10]. Splicing variants account for up to 50% of pathogenic variants in some genes and are suggested to be underrepresented in others due to the lack of their functional analysis at the mRNA level [11,12,13]. Splicing variants could be located anywhere within genes and be occasionally classified as synonymous, missense or nonsense variants. The standard for functional analysis of splicing variants is the analysis of mRNA from patient cells. This approach does, however, have some limitations, e.g., the aberrant mRNA isoform could be degraded by the nonsense-mediated mRNA decay mechanism or the gene of interest may be expressed only in tissue that is difficult to assess. To overcome these issues, the in-vitro minigene splicing assay presents a fast and robust tool for analysis of potentially spliceogenic variants [14,15,16]. 

The need for precise classification of studied variants led to the development of ACMG/AMP guidelines and their gene- and disease-specific refinements curated by ClinGen Workgroups (https://clinicalgenome.org/) (accessed on 15 April 2021) [17,18]. During the analysis of public repositories of PA variants, we faced the problem of incorrect classification of splicing variants, which is inconsistent with ACMG/AMP recommendations. In addition, we revealed a number of variants classified as missense or synonymous, but which were highly spliceogenic according to bioinformatic analysis, and variants classified as splicing but lacking any functional characterization at the mRNA level. 

Therefore, the aim of this work was to improve the PA variant classification, expand the knowledge of splicing variants in the studied genes and evaluate the utility of bioinformatic analysis and the minigene assay for the characterization of PA variants. 

## 2. Materials and Methods

### 2.1. Variant Selection and Bioinformatic Analysis

The subjects of this study were the *PCCA* and *PCCB* gene variants available in the ClinVar and HGMD databases and classified as pathogenic, likely pathogenic and of uncertain significance. In addition, we included four splicing variants previously identified in our lab [19].

Bioinformatic analysis of probable splicing alterations was performed with freely available tools: HumanSplicingFinder 3.1 (HSF3.1) (https://www.genomnis.com/access-hsf) (accessed on 12 June 2020) [20], SpliceAI [21], MMSplice [22], SpiP [23], Ex-Skip (https://ex-skip.img.cas.cz/) (accessed on 15 April 2021) [24], Hexplorer (https://www2.hhu.de/rna/html/hexplorer_score.php) (accessed on 15 April 2021) [25] and HExoSplice (http://bioinfo.univ-rouen.fr/HExoSplice_submit/index.php) (accessed on 15 April 2021) [26]. 

For the variants which create new splicing sites (SS) or disrupt wild type (WT) SSs, the results of HSF3.1, SpliceAI, MMSplice and SpiP were considered. Exonic variants which do not create or disrupt SSs were analyzed using Ex-Skip, Hexplorer and HExoSplice, which are the recommended tools for analysis of probable alteration of exonic regulatory splicing motifs [27]. 

The standard cutoffs and values for the variant to be significant are as follows: HSF3.1 matrices, >15% difference in SS strength; MaxEntScan module of HSF3.1, >30% difference in SS strength; and SpliceAI, delta score > 0.5. MMSplice and SpiP provide the discrete values and corresponding probabilities. For the analysis of exonic regulatory splicing motif alteration, the variants with the highest Ex-Skip, Hexplorer and HExoSplice scores were selected, because this type of prediction has low specificity and depends largely on whether the splicing of the studied exon relies on the recognition of regulatory motifs and their specific positioning [27].

To visualize the probable effect of in-frame deletions caused by exon skipping, the human PCC heterodimer was modeled using SWISS-MODEL (https://swissmodel.expasy.org/) (accessed on 15 April 2021) [28] with the protein databank (https://www.rcsb.org/): 3n6r template (accessed on 15 April 2021). Amino acid conservation was visualized by WebLogo (https://weblogo.berkeley.edu/logo.cgi (accessed on 15 April 2021)) using the *PCCA* and *PCCB* gene trees from Ensemble (ENSGT00940000156083 and ENSGT00940000157741).

### 2.2. Minigene Assay

The *PCCA* and *PCCB* exons with a minimum 100 bp adjacent intronic sequence were amplified by PCR with high fidelity polymerase and cloned into multiple cloning sites between two constitutionally spliced exons of the expression vector pSpl3_Flu2. The pSpl3_Flu2 vector is a modification of the pSpl3_Flu vector (with the deletion of the strong cryptic donor SS downstream of the multiple cloning site), which was used for analysis of splicing variants in the *PAX6* gene [29]. The studied variants were introduced into the minigenes by overlap-extension PCR. Transfection was performed in 24-well plates at ~80% cell confluency with 0.5 μg of plasmid DNA via the calcium phosphate method [30]. After 48 h, cells were harvested, and RNA was extracted and reverse transcribed. Plasmid-specific cDNA was amplified and visualized by polyacrylamide gel electrophoresis (PAGE) with subsequent gel purification (if needed) and Sanger sequencing. 

### 2.3. Variant Calling and Classification

All variants are referred to according to the *PCCA*: NM_000282.4 and the *PCCB*: NM_000532.5 reference sequences and the Human Genetic Variation Society (HGVS) nomenclature (https://varnomen.hgvs.org/) (accessed on 15 April 2021).

The results of the performed functional analysis were combined with published data for the studied variants to precisely classify them according to ACMG/AMP guidelines. The following specific conditions were applied: PS3 (functional studies), alteration of splicing in the vast majority of mRNA molecules that leads to frameshift and nonsense mediated decay (NMD) or predicted deleterious effect at protein level; PP3 (computational data), if more than one of the used bioinformatic algorithms predict splicing alteration; BS3 (functional studies, if a synonymous variant does not cause any significant splicing alterations; and PM3_supportive, if a variant is in trans with other variant of uncertain significance or in a homozygous state in a patient with a phenotype of PA.

## 3. Results

### 3.1. The PCCA and PCCB Variant Selection

Using freely available splicing prediction tools (described in Section 2), we analyzed all pathogenic, likely pathogenic and of-uncertain-significance variants of the *PCCA* and *PCCB* genes available in the HGMD and ClinVar databases, and those previously identified in our lab [19]. From the list of variants for which at least one bioinformatic tool predicted the significant splicing alteration, we excluded those that were already characterized at the mRNA level. Among the variants that were predicted to alter exonic regulatory splicing motifs, we selected eight with the highest predictive scores. 

As a result, 24 variants were selected (the variants that create or alter SSs are presented in Table 1 and the variants that alter regulatory splicing motifs are presented in Appendix A): 13 variants which do not alter the canonical dinucleotides and are predicted to weaken WT SS or create new SSs (*PCCA*: c.183+4_183+7del, c.819+9A>G, c.1187T>G (p.Val396Gly), c.1284+2dup, c.1353+5_1353+9del, c.1643+1dup, c.1899+2_1899+3insCT, c.2040G>A (p.Ala680=), c.2119-9A>G; *PCCB*: c.543G>C (p.Leu181=), c.763G>A (p.Gly255Ser), c.882C>T (p.Pro294=), c.1091-8_1091-3del); 3 variants disrupt canonical dinucleotides (*PCCA*: c.468+1G>A, c.717-2A>G; *PCCB*: c.655-2A>G); and 8 variants were suspected to alter exonic regulatory splicing motifs (*PCCA*: c.431G>T, c.611_613del, c.742G>A, c.893A>G, c.1103A>G, c.1292T>G, c.1367G>T, c.2056G>T). 

### 3.2. Minigene Assay

For the analysis of probable splicing alterations caused by the studied variants, we applied the minigene splicing assay, a fast and effective method for evaluating the spliceogenic effect of genetic variants.

For each studied variant, the corresponding exon with at least 100 b.p. of flanking intronic sequence was cloned into expression vector pSpl3_Flu2 between two constitutionally spliced exons (V1 and V2). Given the large length of flanking introns, 15 exons (the *PCCA* exons 2, 6, 8, 10, 13, 14, 15, 16, 18, 21 and 22 and *PCCB* exons 5, 7, 8 and 11) were cloned separately and only the *PCCA* exon 24 was combined with exon 23 to better reproduce the native genomic milieu, because we detected the splicing artifacts during the *PCCA*: c.2119-9A>G variant analysis. All of the resulting WT minigene constructs demonstrated adequate exon recognition and the absence of cryptic SS activation and, therefore, were used to test the studied variants (Figure 1A). The WT and mutant minigenes were transfected into HEK293T cells. After 48 h, the cells were harvested, and RNA was extracted and reverse transcribed. Minigene-specific cDNA was amplified with primers located in exons V1 and V2 of the vector and visualized by PAGE (Figure 1A). PCR products were Sanger sequenced with preparatory gel extraction if more than one product was visualized.

The results of the minigene assay are presented in Figure 1A,B and Table 2. *PCCA*: c.819+9A>G, which was predicted to activate cryptic SS, and *PCCB*: c.763G>A, c.882C>T, located at the last exonic positions, demonstrated no significant effect on splicing. *PCCA*: c.1187T>G, c.1643+1_1643+2dup, c.2119-9A>G and *PCCB*: c.655-2A>G, c.1091-8_1091-3del led to frameshifting deletions or insertions with no significant amount of the WT mRNA isoform. *PCCB*: c.543G>C, which is located at the last exonic positions and weakens the donor WT SS, causes the in-frame exon skipping with the significant residual amount of full-length mRNA isoform. *PCCA*: c.183+4_183+7del, c.468+1G>A, c.717-2A>G, c.1284+2dupT, c.1353+5_1353+9del, c.1899+2_1899+3insCT, c.2040G>A, and *PCCB*: c.1091-8_1091-3del led to in-frame deletions or exon skipping with no significant amount of WT mRNA isoform. Variants that were predicted to disrupt regulatory splicing motifs (*PCCA*: c.431G>T, c.611_613del, c.742G>A, c.893A>G, c.1103A>G, c.1292T>G, c.1367G>T, c.2056G>T) did not show any difference from the WT (data not shown).

Overall, of the 14 variants that demonstrated the alteration of splicing, only three (*PCCA*: c.1187T>G, c.1643+1_1643+2dup; *PCCB*: c.655-2A>G) led to the complete absence of the WT mRNA isoform and the presence of frameshifted isoforms, which are the substrates of NMD. The remaining variants led to the synthesis of mRNA isoforms which escape NMD. Therefore, to better characterize their deleterious effect, we further analyzed them using PCC enzyme homology modeling.

### 3.3. Analysis of the Affected Protein Structures

To characterize the deleterious effect of the studied variants at the protein level, we performed the homology modeling of the PCC heterodimer and located the functional domains and catalytic residues based on the previously published data [33,34,35] (Figure 2). 

*PCCA*: c.2119-9A>G leads to 8 b.p. frameshifting insertion p.Val707Asnfs*4 with premature stop–codon formation in the *PCCA* last exon. The corresponding *PCCA* mRNA escapes NMD, but the synthesized protein lacks 22 C-terminal amino acids which are involved in the formation of the highly conserved biotinyl-binding domain (Figure 2G). 

The deletion p.His36_Lys61del caused by *PCCA*: c.183+4_183+7del involves the mitochondrial target peptide (amino acids 1–52), thus altering the mitochondrial localization of the PCC.

The remaining in-frame deletions caused by the studied variants involve the highly conserved and, in some cases, catalytic residues of well-characterized domains. As a result, their deleterious effect on protein function is beyond doubt (Figure 2 and Table 2). 

### 3.4. Classification of the Studied Variants

The ACMG/AMP guidelines were used to classify the studied variants [17,18]. The PS3 criterion (well-established in vitro or in vivo functional studies supportive of a damaging effect on the gene or gene product) was applied to the variants which did not demonstrate any significant residual amount of the WT mRNA isoform in the minigene assay and whose deleterious effect was established at the mRNA or protein level. Among these, the variants located in the canonical dinucleotides acquire the PVS1 criterion (null variant) instead. Additionally, we analyzed the available published data for these variants and collected additional criteria to perform their precise classification. As a result, we classified five variants and changed the pathogenic status of nine (Table 2). 

## 4. Discussion

The imprecise classification of identified variants in public databases could confuse researchers and even lead to wrong diagnoses. Recently, the ACMG/AMP guidelines and Clinical Genome Sequence Variant Interpretation Working Groups (ClinGen WGs) provided the most detailed recommendations for standardized variant classification. ClinGen WGs were expanded by gene- and condition-specific expert panels, but recommendations for the precise classifications of splicing variants based on predictive and functional evidence are still being developed. 

The ClinVar and HGMD databases are the main repositories of PA variants. During the detailed analysis of the splicing variants reported there, we faced the problem of unstandardized classification that is not based on any specific assertion criteria, nor supported by the proper functional studies. For example, the *PCCA*: c.1353+5_1353+9del and c.2040G>A variants are reported as “pathogenic”, but the main criteria were the bioinformatic predictions [36]. The *PCCB*: c.1091-8_1091-3del variant is reported as pathogenic, but no functional or any other evidence is provided. Therefore, according to ACMG/AMP guidelines, the proper status of these variants is uncertain significance. The results of our functional analysis clearly established their deleterious effect and, considering the other criteria, allowed us to classify them as pathogenic or likely pathogenic. The *PCCA*: c.2119-9A>G was identified in three PA patients without two causative *PCCA* or *PCCB* mutations, but the authors classified it as a polymorphism because the cDNA analysis showed the normal-sized PCR products (Table 4 of [7]). Our results of the minigene assay suggest that *PCCA*: c.2119-9A>G causes an 8 b.p. insertion in *PCCA* exon 24 and is the spliceogenic loss-of-function variant, which leads to synthesis of the truncated protein with altered biotinyl-binding domain. Because the 8 b.p. insertion could not be distinguished in some cases during gel electrophoresis, and the authors did not report whether Sanger sequencing was performed, their classification is questionable. 

Another widely underestimated nuance is the establishment of the deleterious effect of the variants located in canonical dinucleotides before the application of the PVS1 criteria. The ACMG/AMP recommendations for interpreting the PVS1 criteria state that the researcher should pay attention to potential cryptic SSs, the activation of which could preserve the reading frame. The predictive power of widely used algorithms is still too low to effectively predict the cryptic splice sites’ activation because the additional splicing elements, such as enhancers and silencers, are involved in the specific splice site usage. The analysis should also be performed at the protein level to establish the effect of the potential in-frame indels. For example, the *PCCA*: c.468+1G>A and *PCCA*: c.717-2A>G variants were predicted to cause in-frame deletions, which was confirmed by our minigene assay. Subsequent analysis at the protein level demonstrated that the deletion p.Val139_Leu156del caused by the *PCCA*: c.468+1G>A involves the Tyr143 catalytic residue and clearly established its deleterious effect. *PCCB*: c.717-2A>G leads to the 24 b.p. deletion p.Asp240_Gln247del that alters the beta-sheet in the biotin carboxylase domain and its deleterious effect is not as obvious, although this is the highly flexible and conservative element that serves as the lid for the protein’s active site. 

Missense variants are the most frequent mutations that cause human genetic disease, but their functional characterization at the protein level is a difficult task. When novel rare missense or synonymous variants are found in the studied gene, a researcher should perform bioinformatic analysis with splicing prediction tools, because these variants could disrupt SSs when located at first or last three positions of the exon, activate cryptic SSs or alter motifs of regulatory splicing proteins. If the alteration of splicing is suspected, the patient’s mRNA analysis, or, in its absence, the minigene assay can readily characterize the deleterious effect. The missense variant *PCCA*: c.1187T>G (p.Val396Gly) clearly demonstrated splicing alteration due to cryptic SS activation with complete absence of a full-length mRNA isoform. The synonymous *PCCA*: c.2040G>A (p.Ala680=) variant located in the last exonic nucleotide causes exon skipping in the vast majority of transcripts. By comparison, the *PCCB*: c.543G>C (p.Leu181=) variant leads to exon skipping and a significant residual number of full-length transcripts, thus it could cause a mild phenotype or represent a rare benign variant. Unfortunately, the patient’s data were not described in ClinVar for this variant, so its status remains uncertain due to the lack of sufficient criteria. The *PCCB*: c.882C>T (p.Pro294=) variant demonstrated no significant splicing alterations. This is a rare synonymous variant with no effect either on minigene mRNA splicing or on protein structure, although it could play some more complex roles in molecular pathogenesis, such as modification of mRNA secondary structure or RNA interference. 

The *PCCA*: c.819+9A>G variant was predicted to activate the strong cryptic donor SS, but the minigene assay did not demonstrate any differences from the WT minigene. This could be explained by additional factors involved in splice site recognition that favor WT SS, such as mRNA secondary structure and the specific location of splicing regulatory proteins’ motifs.

The *PCCA*: c.183+4_183+7del, c.468+1G>A and c.1284+2dup, and *PCCB*: c.655-2A>G variants were identified in our lab earlier in patients with the clinical and biochemical phenotype of PA [19]. The performed functional analysis together with other criteria allowed us to classify these as pathogenic variants and establish the patients’ diagnosis.

Variants that were predicted to disrupt regulatory splicing motifs did not show any difference from the WT, although in some genes they account for up to 60% of pathogenic variants [37]. Our results could be explained by the characteristic gene structure, for which the absence of alternative exons and isoforms is described, which suggests the straightforward regulation of splicing, mediated by strong SSs.

Overall, the minigene assay demonstrated a fast and effective approach in the analysis of the *PCCA* and *PCCB* variants. The majority of the presented minigene constructs comprise a single exon and can be easily cloned. Although this approach was not able to detect all of the splicing outcomes, e.g., skipping of two exons or utilization of more distant cryptic SSs, it could be effectively used to estimate whether the identified variant has the spliceogenic potential. 

Regarding the splicing prediction tools, SpliceAI showed the best performance, correctly predicting the effect for 15 of 16 variants. For the variants that disrupt the canonical dinucleotides and are located in the intronic part of SSs, all of the tools correctly predicted the splicing alteration. The most complex variants for analysis showing a significant discrepancy between tools are those located in the exonic part of donor SS (*PCCB*: c.543G>C, c.763G>A, c.882C>T and *PCCA*: c.2040G>A) and those that activate the cryptic SS (*PCCA*: c.819+9A>G, c.1187T>G). 

## 5. Conclusions

The application of the minigene assay for the analysis of the 24 *PCCA* and *PCCB* variants demonstrated a fast and effective approach to establishing their spliceogenic effects. The protein homology modeling could be further used to better characterize the deleterious effect of splicing variants. The results of this work led to the precise ACMG/AMP-based classification of 16 PA variants, for 12 of which we characterized the deleterious effect at the mRNA or protein levels. We suggest that this approach should be expanded to other disease-associated genes to improve the variant classification in mutation databases.

## Figures and Tables

**Figure 1 ijms-22-04154-f001:**
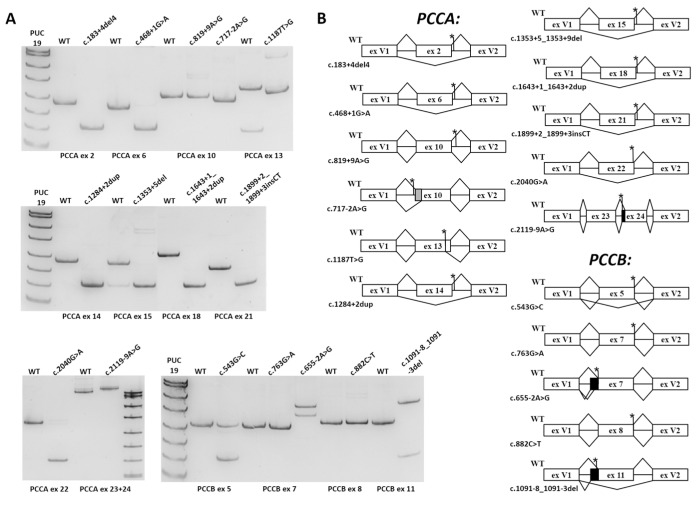
The results of the minigene assay: (**A**) visualization of plasmid-specific PCR products by PAGE; and (**B**) the scheme of splicing patterns for WT and mutant minigenes. Black bars indicate insertion of intron sequences, gray bars indicate deletion of exon sequences and asterisks indicate location of the studied variants. ex, exon.

**Figure 2 ijms-22-04154-f002:**
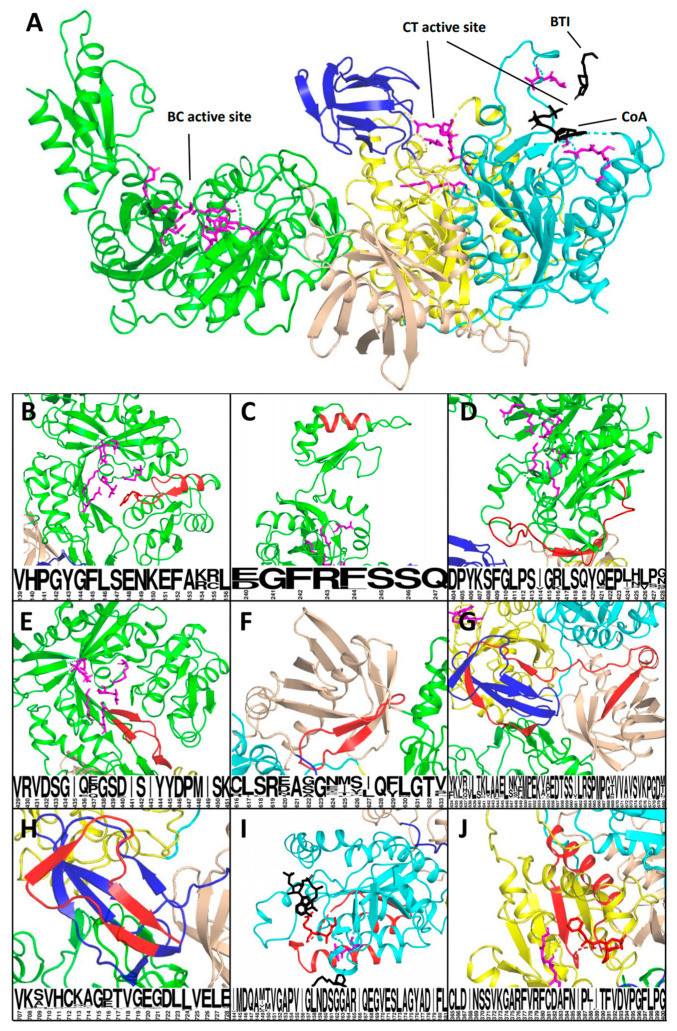
The 3D model of the human propionyl CoA carboxylase (PCC) heterodimer and location of in-frame deletions. The alpha subunit is encoded by *PCCA* and contains the biotin carboxylase (BC) domain (green), the biotin carboxyl carrier protein (BCCP) domain (blue) and the BT domain, which is essential for interactions with the beta subunit (brown). The beta subunit is encoded by *PCCB* and contains the carboxyl transferase (CT) domain (C subdomain in yellow and N subdomain in blue). In the PCC heterododecamer, C and N subdomains of neighbor subunits form the carboxyl transferase active site). Catalytic residues of active sites are indicated in purple. The captions below represent the conservation of deleted amino acids (the higher the letter, the more conservative the corresponding amino acid). (**A**) PCC monomer with concurrent inhibitor BTI and CoA molecules in the active site. (**B**) p.Val139_Leu156del (caused by *PCCA*: c.468+1G>A) involves the BC domain active site and the Tyr143 catalytic residue. (**C**) p.Asp240_Gln247del (*PCCA*: c.717-2A>G) involves the highly conserved beta-sheet of the BC/B subdomain, which serves as the lid of the BC active site. (**D**) p.Asp404_Gly428del (the *PCCA*: c.1284+2dupT) involves the BC domain active site outer wall. (**E**) p.Val429_Lys451del (*PCCA*: c.1353+5_1353+9del) involves the BC domain active site outer wall. (**F**) p.Cys616_Val633del (*PCCA*: c.1899+2_1899+3insCT) involves the BT domain. (**G**) p.Tyr634_Ala680del (*PCCA*: c.2040G>A) involves the flexible stem, connecting BT and BCCP domains. (**H**) p.Val707Asnfs*4 (*PCCA*: c.2119-9A>G) involves the significant part of the BCCP domain. (**I**) p.Ile144_Leu181del (*PCCB*: c.543G>C) involves the CT active site and Gly163, Ala164 and Arg165 catalytic residues. (**J**) p.Cys365_Gly400del (*PCCB*:c.1091-8_1091-3del) involves the CT active site and Phe397, Leu398 and Pro399 catalytic residues.

**Table 1 ijms-22-04154-t001:** The list of studied variants that create or alter SSs only.

Variant Description					Bioinformatic Analysis			
Gene/Exon	Variant	Source	Classified in ClinVar	Predicted Effect	HSF3.0 Matrices/MaxEntScan	SpliceAI DS	MMSplice Result	SPiP (Splicing Disruption Risk)
*PCCA* 2	c.183+4_ 183+7del	Clinvar ID 554669, [19]	US	Alteration of WT DSS	−23.87%/−91.15% for WT DS	0.9 for DL	D	Alter by SPiCE 85.82%
*PCCA* 6	c.468+1G>A	[19]	N/A	Alteration of WT DSS	−28.72%/−78.13% for WT DS	0.99 for DL	D	Alter by SPiCE 85.82%
*PCCA* 10	c.819+9A>G	Clinvar ID 568750	US	Activation of intronic cryptic DSS	+16.82%/+548.07% for CR DS	0.57 for DG	N	Alter by creating de novo splice site, 16.88%
*PCCA* 10	c.717−2A>G	Clinvar ID 850378	LP	Alteration of WT ASS	−30.7%/−69.37% for WT AS	1.0 for AL	D	Alter by SPiCE 98.67%
*PCCA* 13	c.1187T>G (p.Val396Gly)	[31]	N/A	Activation of exonic CR DSS	+44.85%/+447.37% for CR DS	0.96 for DG	N	Alter by creating de novo splice site, 30.18%
*PCCA* 14	c.1284+2dup	[19]	N/A	Alteration of WT DSS	−46.61%/−376.08% for WT DS	0.98 for DL	D	Alter by SPiCE 97.46%
*PCCA* 15	c.1353+5_ 1353 + 9del	Clinvar ID 254165	P based only on bioinformatic analysis (PMID: 27227689)	Alteration of WT DSS	−14.57%/−162.03% for WT DS	0.62 for DL	D	Alter by SPiCE 97.46%
*PCCA* 18	c.1643+1_ 1643+2dup	Clinvar ID 553799	US	Alteration of WT DSS	−13.13%/−70.73% for WT DS	0.74 for DL	D	Alter by SPiCE 85.82%
*PCCA* 21	c.1899+2_ 1899 + 3insCT	[32]	N/A	Alteration of WT DSS	−32.63%/−202.22% for WT DS	1.0 for DL	D	Alter by SPiCE 97.46%
*PCCA* 22	c.2040G>A (p.Ala680=)	Clinvar ID 218256	US, P (based only on bioinformatic analysis (PMID: 27227689)	Alteration of WT DSS	−11.56%/−51.3% for WT DS	0.63 for DL	D	Alter by SPiCE 85.82%
*PCCA* 24	c.2119-9A>G	Clinvar ID 459938	LP	Activation of intronic cryptic ASS	+50.57/+723.95 for CR AS	0.99 for AG	D	Alter by creating de novo splice site, 66.21%
*PCCB* 5	c.543G>C (p.Leu181=)	Clinvar ID 658152	US	Alteration of WT DSS	−12.01/−57.03% for WT DS	0.15 for DL	D	Alter by SPiCE 85.82%
*PCCB* 7	c.763G>A (p.Gly255Ser)	Clinvar ID 557375	US	Alteration of WT DSS	−14.8/−42.95% for WT DS	0.08 for DL	D	Alter by SPiCE 85.82%
*PCCB* 7	c.655-2A>G	[19]	N/A	Alteration of WT ASS	−30.97%/−121.41% for WT DS	1.0 for AL	D	Alter by SPiCE 98.67%
*PCCB* 8	c.882C>T (p.Pro294=)	Clinvar ID 343472	US/LB	Alteration of WT DSS	−2.19/−35.05% for WT DS	0.06 for DL	N	Alter by SPiCE 85.82%
*PCCB* 11	c.1091 -8_1091-3del	Clinvar ID 802010	P, without evidence	Alteration of WT ASS	−5.08%/−71.79% for WT AS	0.86 for AL	D	Alter by SPiCE 66.21%

Variant description: US, uncertain significance; LP, likely pathogenic; P, pathogenic; P*, pathogenic but without any functional evidence; LB, likely benign; WT, wild type; CR, cryptic; ASS, acceptor splicing site; DSS, donor splicing site. Bioinformatic analysis: SpliceAI DS, delta score (indicates the probability of the genomic region to gain or lose the properties of splicing site); AG, acceptor gain; AL, acceptor loss; DG, donor gain; DL, donor loss. Standard cutoffs for bioinformatic tools: HSF matrices, score change > 15%; MaxEntScan, score change > 30%; SpliceAI, delta score > 0.5. The result of MMSplice analysis: D, disease causing variant; N, neutral variant. SPiP represents the results of its incorporated tools and indicates whether the variant alters the splicing.

**Table 2 ijms-22-04154-t002:** Characterization of the Studied Variants’ Effects at mRNA and Protein Levels.

Gene/Exon	Variant	Effect in Minigene/Potential Effect at the *PCCA*/*PCCB* mRNA Level	Potential Effect at the Protein Level	Affected mRNA/Protein Properties	Classified in ClinVar	New Classification, According to ACMG
*PCCA* 2	c.183+4_183+7del	exon skipping/ r.106_183del	p.His36_Lys61del	Mitochondrial target peptide	US	P (PS3, PM2, PM3, PP3, PP4)
*PCCA* 6	c.468+1G>A	exon skipping/ r.415_468del	p.Val139_Leu156del	BC domain active site	N/A	P (PVS1, PM2, PM3, PP4)
*PCCA* 10	c.819+9A>G	no significant effect	unknown	unknown	US	US (PM2, BS3)
*PCCA* 10	c.717-2A>G	24 b.p. deletion/ r.717_741del	p.Asp240_Gln247del	BC/B subdomain	LP	P (PVS1, PM2, PP4)
*PCCA* 13	c.1187T>G (p.Val396Gly)	23 b.p. deletion/ r.1066_1089del	p.Val396Glyfs*37	NMD	N/A	P (PS3, PM2, PM3, PP3, PP4)
*PCCA* 14	c.1284+2dupT	exon skipping/ r.1210_1284del	p.Asp404_Gly428del	BC domain	N/A	P (PS3, PM2, PM3, PP3, PP4)
*PCCA* 15	c.1353+5_ 1353+9del	exon skipping/ r.1285_1353del	p.Val429_Lys451del	BC domain	P*	P (PS3, PM2, PM3_supportive, PP3, PP4)
*PCCA* 18	c.1643+1_ 1643+2dup	exon skipping/ r.1541_1643del	p.Gly514Glufs*9	NMD	US	LP (PS3, PM2, PP3)
*PCCA* 21	c.1899+2_ 1899+3insCT	exon skipping/ r.1846_1899del	p.Cys616_Val633del	BT domain	N/A	LP (PS3, PM2, PP3, PP4)
*PCCA* 22	c.2040G>A (p.Ala680=)	exon skipping/ r.1900_2040del	p.Tyr634_Ala680del	BT and BCCP domain	US, P*	P (PS3, PM2, PM3_supportive, PP3, PP4)
*PCCA* 24	c.2119-9A>G	8 b.p. insertion/ r.2118_2119ins8	p.Val707Asnfs*4	BCCP domain	LP	P (PS3, PM2, PM3, PP3, PP4)
*PCCB* 5	c.543G>C (p.Leu181=)	exon skipping with significant amount of the full-length mRNA isoform/ r.430_543del	p.Ile144_Leu181del	CT domain active site	US	US (PM2, PP3)
*PCCB* 7	c.763G>A (p.Gly255Ser)	no significant effect	unknown	unknown	US	LP (PM1, PM2, PM5, PP2, PP3)
*PCCB* 7	c.655-2A>G	107 b.p. and 56 b.p. insertions/ r.654_655ins107 and r.654_655ins56	p.Asp219Profs*13 and p.Asp219Alafs*32	NMD	N/A	P (PVS1, PM2, PM3, PP4)
*PCCB* 8	c.882C>T (p.Pro294=)	no significant effect	unknown	unknown	US/LB	US (PM2, BS3)
*PCCB* 11	c.1091-8_1091-3del	exon skipping and 96 b.p. insertion/ r.1091_1198del and r.1090_1091ins96	p.Cys365_Gly400del and p.Gly364Glufs*16	CT domain active site and NMD	P*	LP (PS3, PM2, PP3)

Affected mRNA/protein properties: NMD, the resulting mRNA isoform is the substrate of nonsense mediated decay; BC, biotin carboxylase domain; BCCP, biotin carboxyl carrier protein domain; CT, carboxyl transferase domain. Classification; according to ACMG/AMP guidelines: P, pathogenic; LP, likely pathogenic; US, uncertain significance; LB, likely benign.

## Data Availability

The data are submitted to the ClinVar database (https://www.ncbi.nlm.nih.gov/clinvar/ (accessed on 15 April 2021)).

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
