# Peer review of "Functional Analysis of the PCCA and PCCB Gene Variants Predicted to Affect Splicing"

_ijms, 2021, doi:10.3390/ijms22084154_

Round 1

Reviewer 1 Report

Manuscript ID: ijms-1142860

Title: Functional analysis of the PCCA and PCCB genes` variants predicted to affect splicing

Comments to Authors:

In this manuscript, the authors did an important investigation to deepen our knowledge regarding the functional effect of certain SNPs considering the splicing patterns. The approach in this study is required in the field; it helps researchers to understand the functional implications of these variants and more important it takes our understanding to a deeper level rather than just reporting SNPs in the database. The splicing mechanism is a key player in biology and yet very little has been investigated and understood of how it can affect the underlying mechanism of a disease. Investigating 24 reported variants in PCCA and PCCB genes in details will advance the treatment/drug development area of research of propionic acidemia (PA).

I usually send my recommendations for each section separately including the figures and tables, however, in this manuscript, I figured out that I would need to raise my main concerns/recommendations considering the whole manuscript in general in addition to each section. Hoping this will help to get my message across correctly to improve the manuscript.

Some of my main concerns are as follow:

  1. The general concern that I found the manuscript difficult to read. The main concept/idea is clear, at the same time the flow and readability of the text need to improve. In addition, I feel the authors not expressing at length…Not showing the importance of this type of work through the manuscript. I do not see/read the passion of the work, it seems that they are not convinced of what they did, how they expect me to be convinced! This includes most sections of the manuscripts. I see on the journal website that the manuscript is 15 pages, when I downloaded the Pdf I have only 9 pages including references. Am I missing something?

Based on that and considering how essential this type of study is; I  strongly advise to consider rewriting the manuscript to make it better to read, understandable and more importantly, to reveal the importance of this study in the field and make it a good reference for other researchers to use and cite. Adding more details and explain to the reader what has been done clearly in each step is IMPORTANT. Authors cannot expect that other researchers can guess what they did, authors need to present/give the reader the needed details with ease and pleasure. This work covers a very important concept in science, splicing pattern, as a result, we need to pay attention to how to present it clearly and thoroughly...it is as important as performing the experiments. Otherwise, the authors make it very difficult for us (reviewers) to evaluate and assess the work.

  1. Abstract: The abstract reads weak and vague. I would expect to see the top 2 or 3 highlighted results from the study to be mentioned in the abstract. What the authors want the reader to remember mainly from this, what is the take-home message regarding the splicing? It reads like a report or listing facts. Not a good start.
  2. Introduction/ materials and methods: The same concern, authors MUST expand these sections, explain why this work is important. Furthermore, ending the introduction with the motivation to do all this work to set the scene for the rest of the manuscript. I am losing the enjoyment to continue reading it further. It is their work, their idea, their experiments. If they do not explain it with clarity and passion…who will!

Regarding the M&M, very important to draw the authors attention that bioinformatic analysis can change with changing the cut-offs, parameters, etc…..mentioning the details is really important we can go ignore that information in order to help other researchers to follow.

  1. Results/Discussion: for the results, tables and figures MUST be explained in detail and clearly. Very difficult to follow and read. Some abbreviations in the table not clarified. For example, predicted effect abbreviations are similar to SpliceAI scores but they are different. All these small details are IMPORTANT and will make a difference in the quality of the manuscript. I knew it as I do this type of work, others do not. And again, authors don’t expect the reader know what they are talking about and need to go to search for things to understand…it all need to be included in the tables legends and text. Again, figures not explained clearly. Figure legend must be expanded, it is the most important part of the figure. Explain things in the text of the results section too.

Another important point regarding discussion, difficult to extract the take-home messages. No clear and SPECIFIC conclusion. If authors see the need to do further experiments to be convinced with the results, I strongly recommend them to do so…By this, they can be convinced to convince others.

  1. I choose not to include any further comments regarding the experiment design and quality of the work as many details are missing, made it hard to evaluate and judge the work.

I strongly encouraged the author to present/rewrite this work in a different format.

Author Response

Dear Reviewer, we thank you for your comments and agree with them. We made the major revision of our manuscript and expanded and rewrote all of the sections and table captions. Also, following your recommendations, we rewrote the Abstract and Conclusion. The changes are highlighted in red.

Reviewer 2 Report

The manuscript by Bychkov et al reports studies on functional testing of reported splicing variants in the PCCA and PCCB genes for propionic academia. These investigators used mini-gene constructs of PCCA and PCCB genes containing the exon (and its flanking sequence) containing the sequence variants and adjacent exons to test for aberrant splicing. Twenty-four different sequence variants were investigated and re-classified according to pathogenicity. The manuscript adds to knowledge about the genetic etiology of propionic acidemia and points out the weakness of in silico predictions of sequence pathogenicity.

1. The results using mini-gene constructs were appropriately interpreted. But in the absence of intact cells containing the sequence variants, their results may not represent all of the splicing variants that occur in vivo, for example, skipping of two exons or utilization of more distant cryptic splice sites. This should be discussed by the authors so the reader can appreciate the strengths and weaknesses of the mini-gene method.

2. Reading of the text is complicated by the use of many abbreviations, some of which are only defined in the table or figure legends. The authors should define every abbreviation as it is used in the text, or drop certain abbreviations used only once (i.e. LOF, line 150) or a few times. In Table 2, define “BC”, “BCCP” and “B” domains.

3. There is reference to Table S2 (line 86) and Figure 1S (line 125) in the text. Are these typos? I did not see any supplemental material.

4. Typo on line 150: “cases” should be “causes”.54. There is inconsistent capitalization of the titles in the reference section.

Author Response

Dear Reviewer, we thank you for your comments. We made the major revision of our manuscript, expanded and rewrote all of the sections and table captions (as  Reviewer 1 asked for). The changes are highlighted in red. Our answers to your questions are italicized.

1. The results using mini-gene constructs were appropriately interpreted. But in the absence of intact cells containing the sequence variants, their results may not represent all of the splicing variants that occur in vivo, for example, skipping of two exons or utilization of more distant cryptic splice sites. This should be discussed by the authors so the reader can appreciate the strengths and weaknesses of the mini-gene method.

We discussed it in para at line 320.

2. Reading of the text is complicated by the use of many abbreviations, some of which are only defined in the table or figure legends. The authors should define every abbreviation as it is used in the text, or drop certain abbreviations used only once (i.e. LOF, line 150) or a few times. In Table 2, define “BC”, “BCCP” and “B” domains.

We defined every abbrevation.

3. There is reference to Table S2 (line 86) and Figure 1S (line 125) in the text. Are these typos? I did not see any supplemental material.

We moved the Figure 1S to the main text, Table S2 is also submited.

4. Typo on line 150: “cases” should be “causes”.54. There is inconsistent capitalization of the titles in the reference section.

The typo is fixed. References are arranged according to journal`s style.

Round 2

Reviewer 1 Report

Manuscript ID: ijms-1142860_V2

Title: Functional analysis of the PCCA and PCCB genes` variants predicted to affect splicing

Comments to Authors:

In this revised manuscript, the authors did an obvious effort. Glad to see more details and explanation as requested. As I mentioned before how important this study is for the field to deepen our understanding regarding the splicing mechanisms that can underly the disease progression.

The manuscript reads better and clearer. At the same time, there is something missing, in my opinion, the manuscript missing its essence/vitality. I urge the authors to understand and appreciate the importance of the way they present the work. I want them to consider that they are presenting the story of their work, the flow and the feeling of the manuscript will differ, and more importantly, the reader will understand it and follow it better.

Considering the effort authors did in the first round, I encourage them strongly to improve this version from the writing aspect. I will try my best to suggest some tips, and I am really hoping this will help to get my concern clearer to them.

Dear Authors, you did a great job in the first round, all I am asking now is to wrap the manuscript and present it as a story. As a result, the reader will understand it and appreciate it even more. This piece of work is really important and it would shine through if it presented correctly. 

Some of my tips are as follow:

  1. For table 1 legend, the legend still needs expansion. For example, the columns [SpliceAI DS and MMSplice (pathogenicity, efficiency, result)] explain it, make it interesting to understand. Not just reporting. Choose an example or 2 and explain it in detail. The same applies to the next point.
  2. Regarding Figure 1 legend, still very dry. Please expand and give an interesting example, for example, one of the important splicing exons in one of the genes…choose it and expand on it …try to suggest how this affect us biologically, for example, if the splicing can cause the absence of the WT mRNA…how this affects the biology? present suggestions or theories. as a result, when the reader looks at the figure and reads the legend will pick the idea up and can apply it for other examples spontaneously.
  3. Another point, you can explain the example in figure 1 that you did the 3D modelling for and link it with figure 2 …and make this link clear so the reader can see the splicing and its impact on the 3D structure. Make it more interesting to follow.

In summary, link different parts of the papers together smoothly to make a better flow.

Author Response

Dear reviewer, we thank you for your comments. Our answers are italicized and the revised text in the article is indicated in red.

Title: Functional analysis of the PCCA and PCCB genes` variants predicted to affect splicing

Comments to Authors:

In this revised manuscript, the authors did an obvious effort. Glad to see more details and explanation as requested. As I mentioned before how important this study is for the field to deepen our understanding regarding the splicing mechanisms that can underly the disease progression.

The manuscript reads better and clearer. At the same time, there is something missing, in my opinion, the manuscript missing its essence/vitality. I urge the authors to understand and appreciate the importance of the way they present the work. I want them to consider that they are presenting the story of their work, the flow and the feeling of the manuscript will differ, and more importantly, the reader will understand it and follow it better.

Considering the effort authors did in the first round, I encourage them strongly to improve this version from the writing aspect. I will try my best to suggest some tips, and I am really hoping this will help to get my concern clearer to them.

Dear Authors, you did a great job in the first round, all I am asking now is to wrap the manuscript and present it as a story. As a result, the reader will understand it and appreciate it even more. This piece of work is really important and it would shine through if it presented correctly. 

Some of my tips are as follow:

  1. For table 1 legend, the legend still needs expansion. For example, the columns [SpliceAI DS and MMSplice (pathogenicity, efficiency, result)] explain it, make it interesting to understand. Not just reporting. Choose an example or 2 and explain it in detail. The same applies to the next point.

We added the description of Splice AI delta score (lane 150). For MMSplice we deleted the pathogenicity and efficiency values as they are processed further by the tool to give out the discrete values Disease or Neutral and could complicate the interpretation of its result in the absence of recommended cutoffs.

  1. Regarding Figure 1 legend, still very dry. Please expand and give an interesting example, for example, one of the important splicing exons in one of the genes…choose it and expand on it …try to suggest how this affect us biologically, for example, if the splicing can cause the absence of the WT mRNA…how this affects the biology? present suggestions or theories. as a result, when the reader looks at the figure and reads the legend will pick the idea up and can apply it for other examples spontaneously.

We expanded the legend of Figure 1 (lane 197-201).

  1. Another point, you can explain the example in figure 1 that you did the 3D modelling for and link it with figure 2 …and make this link clear so the reader can see the splicing and its impact on the 3D structure. Make it more interesting to follow.

We mentioned it in lane 200.

In summary, link different parts of the papers together smoothly to make a better flow.

Dear Reviewer, we will gladly accept additional tips considering manuscript structure and how to improve the “essence/vitality”, because we think that the current subsections of the Results are passing consistently into each other and present the detailed description of corresponding figures or tables.